# Quantifying Coincidence in Non-Uniform Time Series with Mutual Graph Approximation: Speech and ECG Examples

Piotr Augustyniak [1,*,†] and Grażyna Ślusarczyk [2,†]

1 Department of Biocybernetics and Biomedical Engineering, AGH University of Krakow, 30-059 Krakow, Poland

2 Institute of Applied Computer Science, Jagiellonian University, 30-348 Krakow, Poland; grazyna.slusarczyk@uj.edu.pl

* Correspondence: august@agh.edu.pl

† These authors contributed equally to this work.

**Abstract:** Compressive sensing and arbitrary sampling are techniques of data volume reduction challenging the Shannon sampling theorem and expected to provide efficient storage while preserving original information. Irregularity of sampling is either a result of intentional optimization of a sampling grid or stems from sporadic occurrence or intermittent observability of a phenomenon. Quantitative comparison of irregular patterns similarity is usually preceded by a projection to a regular sampling space. In this paper, we study methods for direct comparison of time series in their original non-uniform grids. We also propose a linear graph to be a representation of the non-uniform signal and apply the Mutual Graph Approximation (MGA) method as a metric to infer the degree of similarity of the considered patterns. The MGA was implemented together with four state-of-the-art methods and tested with example speech signals and electrocardiograms projected to bandwidth-related and random sampling grids. Our results show that the performance of the proposed MGA method is comparable to most accurate (correlation of 0.964 vs. Frechet: 0.962 and Kleinberg: 0.934 for speech signals) and to less computationally expensive state-of-the-art distance metrics (both MGA and Hausdorf: $O(L_1 + L_2)$). Moreover, direct comparison of non-uniform signals can be equivalent to cross-correlation of resampled signals (correlation of 0.964 vs. resampled: 0.960 for speech signals, and 0.956 vs. 0.966 for electrocardiograms) in applications as signal classification in both accuracy and computational complexity. Finally, the bandwidth-based resampling model plays a substantial role; usage of random grid is the primary cause of inaccuracy (correlation of 0.960 vs. for random sampling grid: 0.900 for speech signals, and 0.966 vs. 0.878, respectively, for electrocardiograms). These figures indicate that the proposed MGA method can be used as a simple yet effective tool for scoring similarity of signals directly in non-uniform sampling grids.

**Keywords:** arbitrary sampling; compressed sensing; pattern classification; distance metric; correlation





## 1. Introduction

Time series or patterns are usually defined as uniform. This means á priori knowledge about the pattern is not available except for maximum variability (i.e., frequency in signals). Consequently, the pattern is defined by samples with regular time distribution, which is expected to best reproduce the possible sign of event or change. This frequently used approach uses a common definition of sample value and spacing in all measurements, which results in a vast amount of redundant information, in particular where the event is possible but does not occur [1,2]. The most frequent uniform sampling is thus convenient but neither thrifty nor justified [3].

Indeed, non-uniform sampling occurs often in a general measurement practice. Appearance of unexpected events produces signal jam or distortions and, when in preprocessing, corrupted data series are found partly useless and extracted; a discontinuity occurs

known as a missing data problem. Missing data may also result from measurements not fully controlled in respect of time (i.e., environmental, astronomical, or medical) [4]. Finally, in the case of expensive or long-lasting measurements, one may decide to sample only the most important part of the á priori known part of a data series [5] and leave behind the remaining and reconstruct them (e.g., with interpolation techniques) when necessary [6–8]. The latter approach was a background of fast magnetic resonance imaging sequences (such as FSE/TSE) [9] and gave birth to much wider signal processing mathematics known as compressive sensing [10].

Aside from efficient storage, non-uniform time series are rarely subject to direct data processing [11]. Most applications assume prior projection to a uniform sampling grid where a plethora of well-known methods are available. Lomb transform [12,13], used to estimate the spectrum of non-uniform time series, and Nadaraya–Watson approximation [14], employed to calculate a linear regression in non-equispaced data points, are examples of rare methods for direct processing of non-uniform signals.

Assuming similarity assessment of data streams to be particularly welcome by various classification and decomposition schemes in signal processing, we studied the existing methods of non-uniform signals similarity scoring and proposed a Mutual Graph Approximation (MGA) to address the problem of accurate yet efficient direct comparison of signals. The MGA method does not assume any coincidence of sampling grids and takes uniform sampling as a particular case of non-uniform data acquisition.

The remaining part of this paper is organized as follows. Section 2 provides a review of state-of-the-art non-uniform data similarity scoring. Section 3 presents the original Mutual Graph Approximation method. Section 4 describes experimental validation of all the methods based on two exemplary signals with variable instantaneous bandwidth: the speech signal and the electrocardiogram. Finally, Section 5 provides a discussion of the obtained results and some indications for future research.

## 2. Related Work

Some distance metrics proposed for implementation in similarity assessment of two sections of non-uniformly sampled signals are based on a graph representation of the signal and graph similarity measure. In the graph representation, a limited time series of non-uniform signal values is characterized as consecutive nodes with value and time attributes, and edges with length and slope attributes. Edges and nodes of the signal graph are mutually dependent; thus, edge and node similarity scores are correlated. Kleinberg is considered the first to invent an iterative algorithm determining the graph similarity score [15].

Let $G_A$ and $G_B$ denote two signals to compare. Then, the Kleinberg algorithm iteratively determines the normalized similarity between a given node $i$ in graph $G_A$ and a node $j$ in graph $G_B$, and summarizes the score values for each node of $G_A$. Complementary assessment of similarity is performed for edges of $G_A$ and $G_B$.

Let $x_{i,j}(k)$ stand for similarity of node $i$ in $G_A$ and node $j$ in $G_B$ at stage $k$; then [16]:

$$x_{i,j}(k) = \sum_{\substack{r:(r,i)\in E_A \\ s:(s,j)\in E_B}} x_{r,s}(k-1) + \sum_{\substack{r:(i,r)\in E_A \\ s:(j,s)\in E_B}} x_{r,s}(k-1) \tag{1}$$

where $k$ stands for iteration number and $E_A$, $E_B$ are edge sets in $G_A$ and $G_B$, respectively.

Correspondingly, if $y_{p,q}(k)$ denotes the edge similarity score between edge $p$ in $G_A$ and edge $q$ in $G_B$, then:

$$y_{p,q}(k) = x_{s(p)s(q)}(k-1) + x_{t(p)t(q)}(k-1) \tag{2}$$

where $s(\cdot)$ and $t(\cdot)$ denote the functions assigning source and target nodes, respectively, to edges.

Now, these scores can be represented in a matrix notation as $X_k$ and $Y_k$. Using *source–edge matrix* $A_S$ and *target–edge matrix* $A_T$ as alternative equivalent representation of $G_A$ adjacency structure ($B_S$ and $B_T$ for $G_B$, respectively), one sees that Equations (1) and (2) may be rewritten as

$$
\begin{aligned}
Y_k &\leftarrow B_S^T X_{k-1} A_S + B_T^T X_{k-1} A_T \\
X_k &\leftarrow B_S Y_{k-1} A_S^T + B_T Y_{k-1} A_T^T
\end{aligned}
\tag{3}
$$

where $\leftarrow$ denotes matrix normalization operation required in each step to obtain convergence, and $[\cdot]^T$ is a matrix transposition. Next, following Zager's proposal [16] to transform the matrices into a vector of 'stacked' columns with a $vec(\cdot)$ operator defined as follows:

$$
vec\left(\begin{bmatrix} | & | & \cdots & | \\ v_1 & v_2 & \vdots & v_J \\ | & | & \cdots & | \end{bmatrix}\right) = \begin{bmatrix} v_1 \\ v_2 \\ \vdots \\ v_J \end{bmatrix}
\tag{4}
$$

so, as $y_k = vec(Y_K)$ and $x_k = vec(X_K)$, one can express the iterative update process as

$$
\begin{aligned}
y_k &\leftarrow (A_S^T \otimes B_S^T + A_T^T \otimes B_T^T) x_{k-1} \equiv C x_{k-1} \\
x_k &\leftarrow (A_S \otimes B_S + A_T \otimes B_T) y_{k-1} \equiv C^T y_{k-1}
\end{aligned}
\tag{5}
$$

where $\otimes$ stands for the Kronecker product of matrices, $C$ stands for coupling matrix being the sum of two matrices, each of which has a single '1' entry in each row. Finally, the iterative similarity score $s_k$ may be expressed in the matrix notation as

$$
s_k \equiv \begin{bmatrix} x \\ y \end{bmatrix}_k \leftarrow \begin{bmatrix} 0 & C^T \\ C & 0 \end{bmatrix} \begin{bmatrix} x \\ y \end{bmatrix}_{k-1}
\tag{6}
$$

Assuming the source and terminal nodes in two graphs are similar, which is true in the case of arbitrarily sampled signal strips of the same duration, Zager noticed a coupling between edge and node scores and proposed to simplify the similarity assessment by a single score [16]. It consists of: (A) rewriting Equations (1) and (6) to compute scores for consecutive edges and nodes alternately, (B) expressing edges as difference in consecutive nodes, and (C) reducing node terms accordingly. This approach omits the edges, but, due to the nodes–edges correlation, the similarity remains monotonic. Consequently, the equivalent similarity scores may be assessed with an assignment matrix built for any combination of nodes in both graphs.

Even when referring solely to the graph nodes, we still have to consider the most general case where different arbitrary sampling models are used in both patterns and no correspondence between the sampling intervals is expected. In this scenario, the sampling interval is random and, consequently, each graph node $i \in I$ is attributed with a pair of values of amplitude $v_i$ and time $t_i$. In this pair, the amplitude is discrete-valued, but, unlike in regular digital signals, time is a continuous variable with the only assumption of monotonicity, i.e., $t_i < t_{i+1}$.

Intuitively, the similarity score would be accurate, provided both time series use the same synchronized sampling grids, i.e., $\forall i,j \; t_i = t_j$. Otherwise, the accuracy depends on time difference between corresponding nodes. Introducing a time-related symmetrical envelope $\phi(t)$ to include more influence from the time neighbourhood led to defining a very general but computationally expensive approach that iteratively calculates time difference and multiplies the respective weighting factor through virtually all nodes in $G_A$ and $G_B$.

$$
s = \sum_{i=1}^{I} \sum_{j=1}^{J} d_{i,j} \phi(|t_j - t_i|)
\tag{7}
$$

where $d_{i,j} = \lim_{k\to\infty} x_{i,j}(k)$, $\phi(0) = 1$ and $\lim_{\tau\to\infty} \phi(\tau) = 0$.

In a simplified scenario, for each node in $G_A$, either a time threshold or node number limit may be applied to select the closest (in time coordinate) node in $G_B$ to calculate the distance (in value coordinate). This cumulative approach may also include a weighting factor to modulate the similarity score according to the time attribute of each node. Reduction in complexity is achieved at the price of respective reduction in accuracy, and the time threshold must not be set above the longest sampling interval. Finally, the most simple case of weighting may omit (i.e., set to zero) all values except for the nodes of best time correspondence. This method yields a simple assignment matrix with 'ones' for closest nodes and 'zeroes' elsewhere. This approach proposed in [16] consists of two processes:

- Search for node correspondence minimizing the time difference (admitting one-to-one, but also one-to-many and many-to-one assignments) and
- Calculate the amplitude distance.

A general method for building a graph assignment matrix, partly implemented in our similarity metrics, was proposed in [17]. Accordingly, the example four point assignment matrix $M$ between nodes $i = 1 \ldots 4$ in $G_A$ and nodes $j = 1 \ldots 4$ in $G_B$ may then be denoted as

$$M = \begin{bmatrix} d_{1,1} & 0 & 0 & 0 \\ 0 & d_{2,2} & 0 & 0 \\ 0 & d_{2,3} & 0 & 0 \\ 0 & 0 & d_{3,4} & d_{4,4} \end{bmatrix} \tag{8}$$

In this example assignment, $A(1) \leftrightarrow B(1)$ is of a one-to-one type, nodes $B(2)$ and $B(3)$ were both found the closest (in the meaning of time) neighbours of $A(2)$ (many-to-one assignment), and nodes $A(3)$ and $A(4)$ were both found closest to $B(4)$ (one-to-many assignment). The similarity score $s$ has been proposed as a double sum:

$$s = \sum_{i=1}^{I} \sum_{j=1}^{J} d_{i,j} \tag{9}$$

Another approach to measure the similarity of two shapes given by randomly distributed points is the Hausdorf distance [18,19]. The basic algorithm iterates for all $i = 1 \ldots I$ nodes of $G_A$ to find among $j = 1 \ldots J$ nodes of $G_B$ the closest point $j_i$ to calculate the elementary distance and maintain the maximum of such distances as a similarity score $s$. Thus, for each node of $G_A$, the minimal distance to any node of $G_B$ is computed, and then the maximum of these distances is assigned to $s$

$$s = \max_{i \in G_A} \min_{j \in G_B} d_{i,j} \tag{10}$$

This general definition was proposed for geometric 2D or 3D shapes where all dimensions are unimodal (i.e., given in the same units). In case of signals, however, we always have time $t_i$ as one (independent) dimension and value(s) $v_i$ on the other (dependent) axes. Therefore, the Hausdorf distance modified for signals first takes each node $i = 1 \ldots I$ in $G_A$ to seek closest node $j = 1 \ldots J$ in $G_B$ in the time domain $\min_{j \in G_B}(|t_j - t_i|)$, then calculates their distance $|v_j - v_i|$ in the value(s) domain and maintains the maximum $\max_{i \in G_A}(|v_j - v_i|)$. Thus, for each node of $G_A$, the node of $G_B$ closest in time is found, and then the maximal difference in values for all such pairs of nodes is assigned to $s$. The respective definition is to be rewritten as

$$s = \max_{i \in G_A}\left(\left|v_i - v_{\arg\min_{j \in G_B}(|t_i - t_j|)}\right|\right) \tag{11}$$

The probably weakest point of the original Hausdorf distance definition is its possible asymmetry; i.e., it may happen that $s_{i,j} \neq s_{j,i}$. To mitigate this drawback, a 'bidirectional' Hausdorf distance is calculated as a maximum of two 'one-sided' values.



An interesting alternative metric making use of sample order (i.e., time monotonicity) is the Frechet distance [20]. Its discrete definition [21] is based on nodes $i = 1 \ldots I$ in $G_A$ and nodes $j = 1 \ldots J$ in $G_B$ bound with a set $C_{G_A,G_B}$ of $l = 1 \ldots L$ unique 'coupling leashes' between nodes $a \in G_A$ and $b \in G_B$ in each graph in the following specific way:

- $a_1 = b_1 = 1$
- $a_L = I$ and $b_L = J$
- $(a_{k+1} = a_k \wedge b_{k+1} = b_k + 1) \vee (a_{k+1} = a_k + 1 \wedge b_{k+1} = b_k) \vee (a_{k+1} = a_k + 1 \wedge b_{k+1} = b_k + 1)$, i.e., to create a new leash, at least one node of either graph must advance.

The length of leach $d_{a,b}$ between a pair of nodes $a \in G_A$ and $b \in G_B$ is given as $(|v_a - v_b|)$, and the distance for the particular set of coupling leashes is defined as

$$\|C\| \equiv \max_{l=1\ldots L} \left( |v_{a_l} - v_{b_l}| \right) \tag{12}$$

A particular set $C_{G_A,G_B}$ depends on applied node progress, and a space $\Gamma_{G_A,G_B}$ of all possible coupling sets has to be iterated to find the minimum called Frechet distance:

$$s = \min_{C \in \Gamma_{G_A,G_B}} \|C\| \tag{13}$$

The other methods like SimRank [22], similarity flooding [23], or vertex similarity [24] also use recursively computed similarity scores based on the scores of neighbouring nodes.

Alternative approaches not referring to the distance metrics have also been proposed for non-uniform pattern classification without the need of reconstructing the uniform signals. A Davenport method [25] calculates similarity of compressive measurements and is based on direct inference. Wimalajeewa [26] studies the theoretical background and provides examples of influence from dimensionality reduction in compressed sensing to the classification performance. The later work from the same author compares Bhattacharya distance [27] and Chernoff distance [28] and finds the latter measure as the most robust to data sparsity increase. Wimalajeewa and Varshney [29] also proved the similarity of sampling grids to have great impact on representation of inter-signal correlation in the compressed domain. To minimize this impact, Cleju [30] proposed an arbitrary compressed sensing acquisition matrix based on the nearest correlation between dictionary atoms.

Aside from distance-based clustering models, often yielding quadratic runtime, similarity of patterns can also be expressed without the distance metrics. This group of methods is based on modelling the statistical distribution of patterns and analyzing their content. One of the most widely used examples of such methods is finite mixture models, where the distribution of the variable is represented as a linear mixture of $K$ individually parametrized basic distributions and their contribution coefficients [31]. Also, Latent Class Analysis [32] is based on the combined probability of observing an $x$ value and probability of $x$ being a member of class $k$ [33]. Alternatively grid-based clustering methods partition the data and aggregate them into grids cells [34]. Although they are somewhat based on distance, their great advantage is a significant reduction in the computational complexity, especially for clustering very large datasets. Finally, Gomes et al. presented Regularized Information Maximization, a purely probabilistic framework for discriminative classifier from an unlabeled dataset [35]. Their method is based on a logistic-regression-like model, employs an additional regularization term to control the number of clusters, and does not refer to the notion of distance.

In numerous applications the metric of signal similarity is well represented by its statistical counterpart, i.e., similarity of probability distribution of their values. This approach, known as Kullback–Leibler divergence (KLD, or relative entropy) [36] estimates how much information gathered from one time series is also present in the other. Unfortunately, the KLD is asymmetric and does not satisfy the triangle inequality. Alternatively, the metric obtained as the second derivative of the KLD is widely known as Fisher Information and used to calculate the informational difference between measurements [37]. The other entropy-based method for signal similarity assessment is the mutual information being

the relative entropy of the product of two marginal probability distributions from joint probability distributions [38]. This approach is a measure of the mutual dependence between the two variables; thus, it is frequently used to compare time series like correlation. Probabilistic similarity metrics do not assume regular or irregular sampling; however, their use in non-uniform time series was not found in literature reviews.

## 3. The Mutual Graph Approximation Method

In this paper, it is assumed that considered signals are expressed by non-uniform sample series. A limited time series of each non-uniformly sampled signal is represented in the form of a graph, where nodes representing consecutive samples are connected by edges. Graph nodes are attributed by sample values and time. Such a representation of signals allows us to uniformly encode their characteristic features thanks to the use of attributes. Moreover, this symbolic depiction is useful for signal analysis and classification and allows for comparing our results with the ones of other methods where signals are also described in the form of graphs [39].

To asses a similarity between two sections of sampled signals, a new graph similarity measure that uses the structural similarity of a node neighbourhood specified by a time-based radius is proposed. The pairwise similarity scores for the nodes of two different graphs are determined on the basis of the weighted distance of the node of one graph to the node of the second graph and the time-dependent number of their neighbours. The similarity metrics are applied to nodes representing the time-nearest samples in both graphs.

In our approach, to represent a section of a non-uniformly sampled signal, we consider an attributed graph $G = (V, E, \alpha)$, where $V$ is a finite, nonempty set of nodes, $E \subseteq V \times V$ is a set of edges, and $\alpha : V \rightarrow Value \times Time$ is an attributing function, which assigns value and time attributes to each node. For the sake of simplicity, the value assigned to a node $v$ by $\alpha$ will be denoted as $val(v)$ and the time assigned to it as $t(v)$.

In order to measure the similarity between two series of samples represented as graphs $G_A$ and $G_B$, respectively, the series are aligned with their detection points (Figure 1) and time coordinates for graph nodes are recalculated in reference to this point. Thus, estimation of similarities can be started from the detection point that has zero time coordinate in both graphs and where sample values are directly comparable. At the beginning for each node of graph $G_A$, corresponding node(s) in $G_B$, i.e., with most similar time attribute, are found and the respective assignment matrix is built.

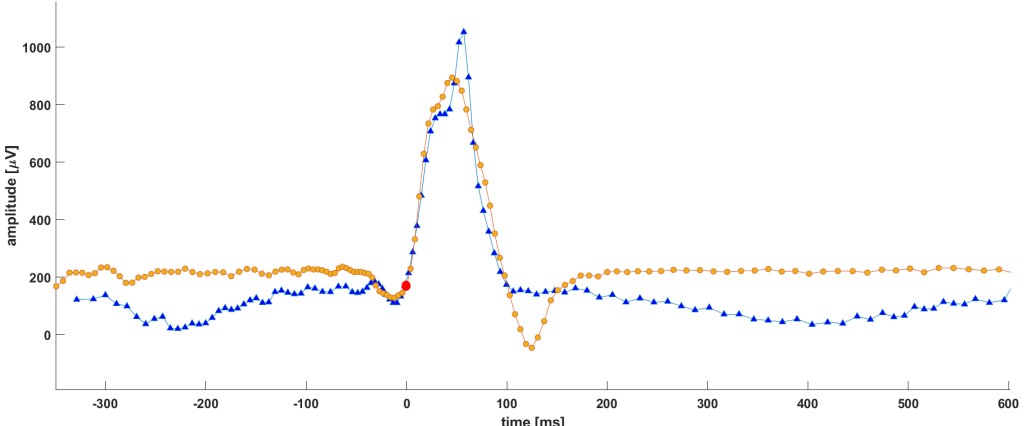

**Figure 1.** An example of two electrocardiogram patterns (orange and blue) with the non-uniform grids (circles and triangles) aligned on their detection point (red) to $t = 0$.

In the next step, values of the second sample series are interpolated in time points determined by the ones assigned to nodes of $G_A$ and added as values of new nodes of $G_B$.

Then, the similarity score between nodes $i$ of $G_A$ and $j$ of $G_B$ representing time nearest samples at $t$ is computed as

$$x_{ij} = |val(i) - val(j)| + \sum_{k \in V_{G_A}:0<|t(k)-t(i)|\leq r} |val(k) - val(s(t(k)))|\phi_i(t(k)), \quad (14)$$

where $s(t(k))$ denotes a node of $G_B$ with $t(s) = t(k)$, $\phi_i(t(k))$ is a weighting factor depending on the time distance between nodes $i$ and $k$, and $r$ is the time specifying the radius of the neighbourhood nodes of $i$ taken into account.

It can be noted that the absolute differences between values of nodes representing samples taken or computed by interpolation at the same time are weighted against the time distance from node $i$. In this paper, we assume that the coefficient $\phi$ is proportional to the time distance in a linear way, and it is computed as $\phi_i(t(k)) = \frac{1}{|t(i)-t(k)|}$, where $i \neq k$.

In Figure 2, the principle of the similarity score calculation for non-uniform patterns is explained. In the examples presented in Figure 2c,d, eight neighbours of the given nodes in $G_A$ and $G_B$, that fall in the time interval of 20 ms around these nodes are considered.

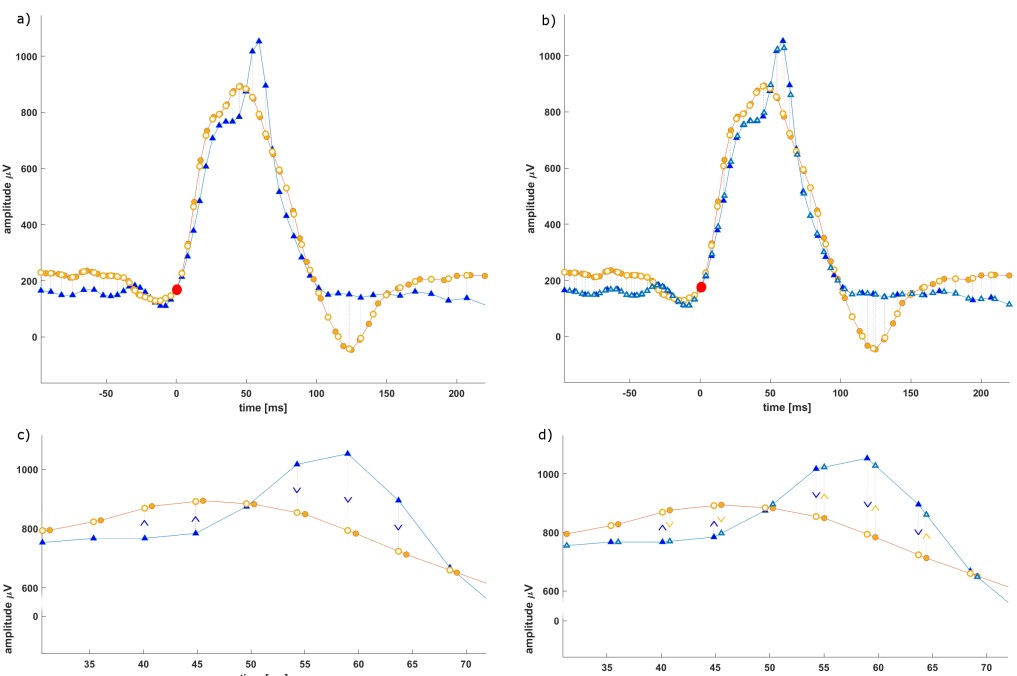

**Figure 2.** Calculating the similarity score with the Mutual Graph Approximation; solid circles are original samples of $G_A$, solid triangles are original samples of $G_B$, (**a**) approximation of $G_A$ values in time points where samples in $G_B$ occur (empty circles) allows to calculate distance $G_B$ to approximated $G_A$, (**b**) approximation of $G_B$ values in time points where samples in $G_A$ occur (empty triangles) allows to calculate distance $G_A$ to approximated $G_B$, (**c**) details of (**a**) in the time range 30–70 ms, (**d**) details of (**b**) in the time range 30–70 ms; '∧' and '∨' marks points where approximated values of the target graph were calculated according to the sampling grid of source graph.

When all pairwise similarity scores between nodes of $G_A$ and $G_B$ are summed up, then the same process is repeated for graph $G_B$; i.e., samples of the signal represented by $G_A$ are interpolated in time points determined by times assigned to nodes of $G_B$, and pairwise similarity scores between nodes of $G_B$ and $G_A$ are added up. This operation justifies referring to the proposed procedure as Mutual Graph Approximation (MGA). The average value of the sum of the obtained distances between $G_A$ and $G_B$, and between $G_B$ and $G_A$ is the similarity score and makes the background for decision, e.g., of the class membership.

It is noteworthy that, instead of iterating through all nodes of $G_B$ for each node of $G_A$, we only have to scan once through all points in $L_1$ and $L_2$ (i.e., the computational

complexity is $O(L_1 + L_2)$). In case the source node is the synchronization point, time coordinates are identical, so no further search is necessary. For any other source point, the search is based on time precedence of samples. That means for a given pair $G_A(t_i, v_i)$ and $G_B(t_j, v_j)$, i.e., source and target points closest in time, the algorithm advances by one sample to next source point $G_A(t_{i+1}, v_{i+1})$ and starts searching for next target point from $G_B(t_j, v_j)$ for the minimum of $|t_{i+1} - t_{j+k}|$. Due to time monotonicity, a single minimum is expected. In case of similar grids, number of search attempts (i.e., $k$ points necessary to find the minimum) will be similar for $L_1$ and $L_2$. However, when $L_1$ and $L_2$ differ (e.g., $L_1 < L_2$, i.e., $G_A$ is sparser than $G_B$), fewer source points $L_1$ in $G_A$ and more target points $L_2$ in $G_B$ have to be checked for being closest in time.

## 4. Experimental Validation

The experiment aimed to investigate the performance of the proposed distance measure in a practical task of signals classification. Two signal types most represented in signal processing literature have been selected as working examples: speech signal and the electrocardiogram. To maintain the generality, we assume not using any speech- or cardiac-specific signal processing procedures. Consequently, although not optimal with regard to these two signals, the procedures may easily be applied to other kinds of signals. The goal of the experiment was to answer the following questions:

1.　What is the performance of the proposed MGA method with regard to other state-of-the-art distance metrics?
2.　Can direct comparison of non-uniform signals be an equivalent for correlation in applications such as signal classification?
3.　What is the role of a bandwidth model in maintaining the performance of non-uniform distance metrics as classification criteria?

### 4.1. Selection and Preprocessing of Test Signals

As an example, speech data collection of the Manitoban speech dataset was selected [40]. The database was issued as reference for testing speech segmentation and distinctions and contains 44 recordings of phonem's alphabet made at 44.1 kHz. The subset used in the experiment included 5 most common alphabet examples: {'28'-'bad', '7'-'church', '32'-'book', '26'-'bid', and '12'-'other'}. The length of each signal has been limited to 2 s.

As an example of the ECG data collection, the MIT-BIH Arrhythmia Database was selected [41]. It consists of 48 half-hour examples of the most frequent arrhythmia recorded in two channels at 360 Hz. The subset used in the experiment included 20 s of medically homogeneous single-channel strips extracted out of 5 most common arrhythmia examples: {'1'-*Normal*, '11'-*Ventricular*, '8'-*Atrial premature beat*, '3'-*Right bundle branch block beat*, and '13'-*Fusion of ventricular and normal beat*}.

These two sources provided 5 signals each of the length $T$ of either 88200 samples or 7200 samples, respectively. In both cases, we assumed the average local bandwidth to be equal a quarter of the Nyquist frequency. The reference, uniform signals are denoted as $Sig_{U\#}$, where '#' is the numerical label (i.e., {'28', '7', '32', '26', '12'} and {'1', '11', '8', '3', '13'}, respectively) from the original dataset.

In order to calculate the local bandwidth of each signal $Sig$ with sampling interval $SI$, the Short-Time Fourier Transform was first applied to uniform time series to produce the spectrogram $Sgrm(t, f)$ in a Gaussian window of 8 samples. Each time slot of the spectrogram was then analyzed to detect the frequency point $F(t)$ cutting out a given percentage $Tsh$ of local spectral energy. Finally, the time series of local points $bw(t)$ were averaged to produce the local bandwidth estimate $BW(t)$.

% local bandwidth estimate
```
for t = 0 : T − 8 : step = 8 do
    Sgrm(t, f) = STFT(Sig(t, t + 8))
    bw(t) = F(t) : sum(Sgrm(t, f > F(t))) <= Tsh * sum(Sgrm(t, f))
```

```
    end for
    for t = 0 : T − 8 : step = 8 do
        BW(t) = (bw(t − 8) + bw(t) + bw(t + 8))/3
        SIraw(t) = SI/BW(t)
    end for
```

The series of sampling intervals $SIraw(t)$ is now ready to determine the time of sampling points for non-uniformly sampled version of each signal denoted as $Sig_{N\#}$. However, to investigate the role of the local bandwidth, we also produced a random (i.e., not regarding the local bandwidth) version of each signal denoted as $Sig_{R\#}$. To maintain the same average value and value distribution of random non-uniform sampling, we simply applied random sampling without replacement to the $SIraw(t)$ series and created a new series $SIrnd(t)$ with the same interval values as in $SIraw(t)$ but in a random order.

Next, the calculation of new sampling grids was performed with the same procedure to $SIraw(t)$ and $SIrnd(t)$.

```
% calculating the non-uniform sampling grid
    CSigLen = 0       % cumulated signal lenght
    k = 0             % running number of the sampling interval
    SPoint(k) = 0     % the first sample of uniform and non-uniform grids falls at t = 0
    while CSigLen < T do
        k = k + 1
        SPoint(k) = CubicSpline(SIraw(t), SPoint(k − 1))
        CSigLen += SPoint(k)
    end while
    K = k             % total number of irregular sampling intervals
```

And, finally, we come to non-uniformly sampled signal values:

```
% calculating signal values on non-uniform grid
    for k = 0 : K do
        Val(k) = CubicSpline(Sig, SPoint(k))
    end for
```

to represent each signal $Sig_{N\#}$ or $Sig_{R\#}$ as a sequence of non uniform samples: {$Val(k)$, $SPoint(k)$}.

The same resampling algorithm was used to restore the non-uniform signal values back to the uniform sampling grid. In this case, however, the sampling grid was defined by a constant time delay equivalent to sampling interval of respective original file and the cubic spline interpolation was applied to approximate values given at non-uniform grid in the equally spaced time points. The restored signals are denoted as $Sig_{BN\#}$ when restored from the bandwidth-related non-uniform grid and $Sig_{BR\#}$ when restored from the random non-uniform grid.

### 4.2. Experiment Setup

The experiment was performed with 5 described distance metrics for each of 3 versions {'uniform', 'bandwidth-related non-uniform', and 'random non-uniform'} of 10 signal strips. The calculation of signal distance was made for several iterations with constant time-shift $\triangle t$, which mimics the definition-based calculation of cross-correlation function. It is noteworthy that, in case of $Sig_{U\#}$, $Sig_{BN\#}$, and $Sig_{BR\#}$ with the uniform sampling grid, time shift means simply advancing sample indexes, whereas, in case of $Sig_{N\#}$ and $Sig_{R\#}$ with non-uniform sampling grid, the operation lies in adding $\triangle t$ to time coordinate of every sample.

Provided the uniform sampling grid may be considered as the particular case of the non uniform one, signals sampled at different grids may also be compared. The only operation needed is of purely technical nature and lies in representing regular samples in the form $\{Val(k), SPoint(k)\}$.

Within the experiment, distances between signals were calculated separately in the group of speech signals and cardiac signals. For reference, we started with original signal similarity score and restored signal similarity score to see how far resampling affects the signal correlation. For this purpose, a regular cross-correlation was applied as first-hand measure of the distance.

Secondly, we measured the signal distance with use of implemented methods presented in Sections 2 and 3. All the measures have been equivalently applied to uniform and non-uniform sampling grid signals. In tests with the MGA method, the radius specifying the time around node $i$ taken into account (i.e., the time span of function $\phi$, see Equation (14)) was set to 0.5 ms and 100 ms for the speech signals and the electrocardiograms, respectively, in order to include at least two samples of the signal. This part of experiment revealed performance of all measures in the role of signal classification criteria.

Finally, we compared the non-uniform signal distance measured with all tested metrics between the bandwidth-related and random non-uniform sampling grids (i.e., $Sig_{N\#}$ and $Sig_{R\#}$). The idea here was to demonstrate that the non-uniform sampling model related to the local bandwidth plays an important role in preserving distinctive features of the signal.

*4.3. Experiment Data Post Processing*

Determining the detection points in real signals may not be an easy and repeatable task, in particular when in non-uniform series the point usually falls between samples. To mitigate this potential source of error and make the evaluation independent, we simulate the misalignment of these points. To this aim all measures have been calculated as correlation-type functions i.e., representing the signal distance as dependent on time-shift $\triangle t$. In the classification task, however, only the minimum distance, corresponding to maximum similarity is taken into account. In the presentation of results hereafter, this 'best matching' value was also maintained as the only estimate of the distance. The distance calculated by some of the implemented algorithms return values dependent on signal amplitude, length and sample count. Therefore, to compare all methods adequately, we normalize the similarity score to the range {0, 1}, where, similarly to the reference cross-correlation, '0' means the signal distance is infinite and similarity is so little as not measurable and '1' means the signal distance is equal to zero and signals are perfectly similar or identical. In case of Hausdorf [18] and Frechet [20] methods, a distance value being maximum or minimum of the set is therefore independent on the set power and the normalized similarity score $ns$ may be calculated as $ns = \frac{(Fs-s)}{Fs}$, where $Fs$ is the full scale amplitude, i.e., maximum span of two-side boundary of both signals. In case of Kleinberg [15] and Zager [16] scores, distances are defined as sums and to be independent of the series length need to be divided by total number of nodes $L_1 + L_2$. The proposed MGA also belongs to this category. The normalized similarity score is then calculated as $ns = \frac{Fs - \frac{s}{L_1+L_2}}{Fs}$, where $L_1$ and $L_2$ are point numbers in respective non-uniform time series.

*4.4. Experiment Results*

Table 1 presents the results of the similarity score expressed by the maximum of the cross-correlation function in pairs $Sig_{U\#}$ and $Sig_{U\#}$. Note that the 'ones' on diagonals mean perfect self-similarity of each signal.

**Table 1.** Results of the similarity score expressed by maximum of the cross-correlation function for reference uniform signals.

| | | | Speech Signals | | |
|---|---|---|---|---|---|
| # | '28' | '7' | '32' | '26' | '12' |
| '28' | 1 | 0.87 | 0.65 | 0.48 | 0.41 |
| '7' | | 1 | 0.43 | 0.51 | 0.63 |
| '32' | | | 1 | 0.38 | 0.57 |
| '26' | | | | 1 | 0.72 |
| '12' | | | | | 1 |
| | | | Electrocardiograms | | |
| # | '1' | '11' | '8' | '3' | '13' |
| '1' | 1 | 0.33 | 0.89 | 0.81 | 0.67 |
| '11' | | 1 | 0.41 | 0.39 | 0.48 |
| '8' | | | 1 | 0.69 | 0.61 |
| '3' | | | | 1 | 0.58 |
| '13' | | | | | 1 |

Table 2 presents the results of the similarity score expressed by the maximum of the cross-correlation function in pairs $Sig_{U\#}$ and $Sig_{BN\#}$.

**Table 2.** Results of the similarity score expressed by maximum of the cross-correlation function for reference uniform signals and uniform signals restored from the bandwidth-related non-uniform grid.

| | | | Speech Signals | | |
|---|---|---|---|---|---|
| # | '28' | '7' | '32' | '26' | '12' |
| '28' | 0.96 | 0.83 | 0.63 | 0.43 | 0.40 |
| '7' | | 0.97 | 0.42 | 0.49 | 0.61 |
| '32' | | | 0.96 | 0.37 | 0.55 |
| '26' | | | | 0.95 | 0.68 |
| '12' | | | | | 0.96 |
| | | | Electrocardiograms | | |
| # | '1' | '11' | '8' | '3' | '13' |
| '1' | 0.94 | 0.32 | 0.85 | 0.81 | 0.64 |
| '11' | | 0.94 | 0.40 | 0.39 | 0.43 |
| '8' | | | 0.96 | 0.67 | 0.60 |
| '3' | | | | 0.98 | 0.55 |
| '13' | | | | | 0.97 |

While Table 1 is an absolute reference of mutual similarity of test signals, the data in Table 2 show how discrimination power of the calculated distance decreases with resampling of signals to the bandwidth-related non-uniform grid.

Table 3 presents the results of the similarity score expressed by the maximum of the cross-correlation function in pairs $Sig_{U\#}$ and $Sig_{BR\#}$.

**Table 3.** Results of the similarity score expressed by maximum of the cross-correlation function for reference uniform signals and uniform signals restored from the random non-uniform grid.

| Speech Signals | | | | | |
|---|---|---|---|---|---|
| # | '28' | '7' | '32' | '26' | '12' |
| '28' | 0.91 | 0.87 | 0.65 | 0.48 | 0.41 |
| '7' | | 0.90 | 0.40 | 0.43 | 0.60 |
| '32' | | | 0.88 | 0.32 | 0.51 |
| '26' | | | | 0.92 | 0.66 |
| '12' | | | | | 0.89 |
| Electrocardiograms | | | | | |
| # | '1' | '11' | '8' | '3' | '13' |
| '1' | 0.89 | 0.30 | 0.82 | 0.74 | 0.61 |
| '11' | | 0.83 | 0.35 | 0.34 | 0.42 |
| '8' | | | 0.85 | 0.63 | 0.55 |
| '3' | | | | 0.91 | 0.50 |
| '13' | | | | | 0.91 |

The data in Table 3 show decreased discrimination power of calculated distance due to resampling of signals to the random non-uniform grid. This result shows the role of applying the sampling grid adequate to local signal features.

Below are the results produced by tested distance metrics obtained with non-uniform signals on the bandwidth-related grid. This choice was justified by plausible practical application; however, uniform signals and non-uniform signals on the random grid may also be tested this way. The data in Table 4 show signals similarity scores based on Kleinberg distance calculated according to its original definition [15].

**Table 4.** Results of the similarity score expressed by maximum normalized Kleinberg distance with non-uniform signals on the bandwidth-related grid.

| Speech Signals | | | | | |
|---|---|---|---|---|---|
| # | '28' | '7' | '32' | '26' | '12' |
| '28' | 0.94 | 0.82 | 0.62 | 0.42 | 0.40 |
| '7' | | 0.94 | 0.40 | 0.48 | 0.59 |
| '32' | | | 0.93 | 0.35 | 0.52 |
| '26' | | | | 0.92 | 0.65 |
| '12' | | | | | 0.94 |
| Electrocardiograms | | | | | |
| # | '1' | '11' | '8' | '3' | '13' |
| '1' | 0.93 | 0.30 | 0.81 | 0.80 | 0.63 |
| '11' | | 0.93 | 0.39 | 0.38 | 0.43 |
| '8' | | | 0.94 | 0.62 | 0.57 |
| '3' | | | | 0.94 | 0.52 |
| '13' | | | | | 0.92 |

The data in Table 5 show signals similarity scores based on Zager distance according to [16].

**Table 5.** Results of the similarity score expressed by maximum normalized Zager distance with non-uniform signals on the bandwidth-related grid.

| Speech Signals | | | | | |
|---|---|---|---|---|---|
| # | '28' | '7' | '32' | '26' | '12' |
| '28' | 0.94 | 0.81 | 0.62 | 0.41 | 0.38 |
| '7' | | 0.93 | 0.39 | 0.43 | 0.56 |
| '32' | | | 0.91 | 0.32 | 0.50 |
| '26' | | | | 0.91 | 0.63 |
| '12' | | | | | 0.90 |
| Electrocardiograms | | | | | |
| # | '1' | '11' | '8' | '3' | '13' |
| '1' | 0.91 | 0.30 | 0.83 | 0.78 | 0.61 |
| '11' | | 0.89 | 0.36 | 0.35 | 0.42 |
| '8' | | | 0.90 | 0.63 | 0.55 |
| '3' | | | | 0.91 | 0.50 |
| '13' | | | | | 0.91 |

The data in Table 6 show signals similarity scores based on Hausdorf distance [18]. To mitigate possible asymmetry, 'bidirectional' Hausdorf distance is calculated as a maximum of two 'one-sided' values.

**Table 6.** Results of the similarity score expressed by maximum normalized Hausdorf distance with non-uniform signals on the bandwidth-related grid.

| Speech Signals | | | | | |
|---|---|---|---|---|---|
| # | '28' | '7' | '32' | '26' | '12' |
| '28' | 0.92 | 0.80 | 0.60 | 0.41 | 0.37 |
| '7' | | 0.93 | 0.37 | 0.47 | 0.57 |
| '32' | | | 0.92 | 0.33 | 0.52 |
| '26' | | | | 0.92 | 0.63 |
| '12' | | | | | 0.91 |
| Electrocardiograms | | | | | |
| # | '1' | '11' | '8' | '3' | '13' |
| '1' | 0.91 | 0.27 | 0.81 | 0.80 | 0.62 |
| '11' | | 0.92 | 0.38 | 0.37 | 0.42 |
| '8' | | | 0.91 | 0.60 | 0.53 |
| '3' | | | | 0.92 | 0.50 |
| '13' | | | | | 0.92 |

The data in Table 7 show signals similarity scores based on Frechet distance implemented according to [20].

**Table 7.** Results of the similarity score expressed by maximum normalized Frechet distance with non-uniform signals on the bandwidth-related grid.

| Speech Signals | | | | | |
|---|---|---|---|---|---|
| # | '28' | '7' | '32' | '26' | '12' |
| '28' | 0.96 | 0.88 | 0.61 | 0.45 | 0.40 |
| '7' | | 0.97 | 0.39 | 0.49 | 0.61 |
| '32' | | | 0.95 | 0.35 | 0.55 |
| '26' | | | | 0.96 | 0.68 |
| '12' | | | | | 0.97 |
| Electrocardiograms | | | | | |
| # | '1' | '11' | '8' | '3' | '13' |
| '1' | 0.94 | 0.30 | 0.85 | 0.82 | 0.63 |
| '11' | | 0.94 | 0.41 | 0.40 | 0.45 |
| '8' | | | 0.93 | 0.63 | 0.57 |
| '3' | | | | 0.94 | 0.53 |
| '13' | | | | | 0.94 |

The data in Table 8 show signals similarity scores based on the proposed Mutual Graph Approximation distance.

**Table 8.** Results of the similarity score expressed by maximum normalized Mutual Graph Approximation distance with non-uniform signals on the bandwidth-related grid.

| Speech Signals | | | | | |
|---|---|---|---|---|---|
| # | '28' | '7' | '32' | '26' | '12' |
| '28' | 0.95 | 0.85 | 0.61 | 0.44 | 0.40 |
| '7' | | 0.97 | 0.40 | 0.50 | 0.62 |
| '32' | | | 0.95 | 0.36 | 0.55 |
| '26' | | | | 0.98 | 0.67 |
| '12' | | | | | 0.97 |
| Electrocardiograms | | | | | |
| # | '1' | '11' | '8' | '3' | '13' |
| '1' | 0.95 | 0.32 | 0.85 | 0.82 | 0.64 |
| '11' | | 0.94 | 0.40 | 0.39 | 0.43 |
| '8' | | | 0.95 | 0.64 | 0.59 |
| '3' | | | | 0.97 | 0.54 |
| '13' | | | | | 0.97 |

Table 9 summarizes the key performance parameters and computational complexity of all the tested methods. To measure the metric performance, we focused on the reproducibility of the similarity between non-uniform time series, neglecting the discrimination power. This simplifying assumption is justified by the fact that we have a reference of perfect similarity, which is signal self-similarity, while we do not actually have a good reference of dissimilarity. To calculate the metric performance, all diagonals of respective confusion matrices have been averaged. The average and standard deviation of normalized similarity scores are accompanied by computational complexity estimates.

**Table 9.** Performance parameters and computational complexity of the tested methods; best results are marked in bold; $L_1$ and $L_2$ are sample numbers of two signals; please note that sample numbers covering the equivalent time in both signals may be different.

| Method | Speech Signals | | Electrocardiograms | | Complexity |
|---|---|---|---|---|---|
| | avg. | std. | avg. | std. | |
| Kleinberg [15] | 0.934 | 0.0089 | 0.932 | 0.0084 | $O(L_1 \times L_2)$ |
| Zager [16] | 0.918 | 0.0164 | 0.904 | 0.0089 | $O(L_1 + L_2)$ |
| Hausdorf [18] | 0.920 | **0.0071** | 0.916 | 0.0055 | $O(L_1 + L_2)$ |
| Frechet [20] | 0.962 | 0.0084 | 0.938 | **0.0045** | $O((L_1 + L_2)^2)$ |
| MGA (this work) | **0.964** | 0.0134 | **0.956** | 0.0134 | $O(L_1 + L_2)$ |

## 5. Discussion

The experiment provided a practical proof of concept for the proposed Mutual Graph Approximation as an effective tool for comparing the similarity of signals in the non-uniform sampling grids. In general, there is no assumption about sampling grids correspondence; thus, the regular grid was considered as the particular case of the non-uniform grid. Three questions motivating the experiment were answered as follows:

1. The performance of the proposed MGA method is comparable to most accurate and to less computationally expensive state-of-the-art distance metrics.
2. Direct comparison of non-uniform signals can be equivalent to cross-correlation of resampled signals in applications such as signal classification in both accuracy and computational complexity.
3. The bandwidth model drives the resampling process, which is potentially the primary cause of inaccuracy. An inadequate non-uniform representation leads to erroneous distance estimation.

The graph representation of non-uniformly sampled signals uses only simple structures and seems oversized to non-uniform signals. It is noteworthy that such an approach paves the way to studying the similarity of more complicated structures, such as shapes in images given by a set of non-uniform samples.

### 5.1. Metrics Performance Accuracy vs. Complexity

Analysis of the data in Table 9 leads to the statement that two more accurate state-of-the-art methods (Frechet and Kleinberg) at the same time show higher computational complexity. Two other methods (Hausdorf and Zager) are less complex, but also less accurate. In this context, the proposed a method based on Mutual Graph Approximation that shows the accuracy comparable to the best (Frechet) method, while its complexity remains only slightly higher than that of the Hausdorf method. The last two rows of Table 9 show very close average values of Frechet and MGA in the case of speech signals; however, lower standard deviation is the advantage of Frechet. In the case of electrocardiograms, the superiority of MGA is clearer, but, again, lower standard deviation advocates for Frechet.

The importance of computational complexity stems from virtually very frequent use of similarity score, being a non-uniform equivalent of cross-correlation, in a wide range of signal processing procedures. These include non-uniform pattern analysis and classification, direct decomposition on bases and frames (i.e., non-uniform to time-scale transform), principal component analysis, two-channel analysis, and many others.

To maintain low computational complexity, we used a linear approximation of new values of $G_A$ in timepoints given by $G_B$ samples (and vice versa). Alternative weighting (i.e., functions $\phi_i(t(k))$) is also possible and should be studied in the future as a possible trade-off between the computational complexity and the accuracy of the metrics.

### 5.2. Comments on Experimental Results

In the experiment, we aimed to face all the problems related to the implementation of the new MGA method but also to show how to compare signals in their non-uniform and unrelated grids.

We showed that identical signals (Table 1) can also be considered very similar when projected to a non-uniform grid and backwards as far as the grid is selected according to the local bandwidth (Table 2). It should be noticed that forward and backward projection both use cubic spline interpolation; hence, bit-accurate reconstruction of signals on a regular grid was not expected as a result. Applying an inadequate sampling grid (such as a random non-uniform grid in our example) leads to data loss and affects the similarity score (Table 3). Thanks to usage of the same collection of sampling intervals in sequential and random order, we had identical data throughput of both $Sig_{N\#}$ and $Sig_{R\#}$ time series, and, consequently, the data in Table 2 (the bandwidth-related non-uniform grid) and Table 3 (the random non-uniform grid) may be directly compared. It is also noteworthy that the non-uniform sampling grid is individually determined based on instantaneous bandwidth calculated for a given pattern. Consequently, different patterns will also differ in time points where the signal value is picked.

It may be noticed in Tables 4–8 that 'Speech Signals' show higher values of most similarity scores than 'Electrocardiograms'. This is a result of applying the same rules ($Tsh = 0.99$) of the local bandwidth detection for two signals of different natures. The spectrum of speech signals reaches 99% of its energy much further from its Nyquist frequency than the spectrum of electrocardiograms. ECG records taken from other databases (e.g., with sampling frequency of 500 Hz) or speech signals first decimated to 22050Hz will alter this relationship.

### 5.3. Limitations of the Test Signals

Although our goal was to present the Mutual Graph Approximation as a general similarity score for virtually all non-uniform signals, in the experiment, we employed selected speech signals and electrocardiograms with consideration of their limitations.

Each speech sample represents one vowel sounding within a given word. The words used are of different length and were recorded from a speaker of particular vocal features (e.g., male or female). In the case of speech signals, we did not find literature references to non-uniform sampling proposals.

The normal electrocardiogram is usually considered as quasi-periodic. In the pathological record, a homogenous strip (i.e., continuously representing a given pathology) had to be selected and extracted. There are several proposals of so called 'compressive sampling' of ECGs based on the statistical approach [42] and a single proposal of the perception-based non-uniform sampling model by Augustyniak [43].

To demonstrate the universal applicability of the MGA metrics, we intentionally avoided speech- or electrocardiogram-specific signal processing. Consequently, the local bandwidth calculated in both cases from the local energy of the spectrogram may not be optimal in each case. Moreover, we assume no important signal components are present above the local bandwidth limit [44]. Adopting the white noise model for the ultra-bandwidth content, we found that aliasing of noise produces a sub-bandwidth noise component, and, for the sake of simplicity, we neglected a tunable anti-aliasing filter that normally is required when resampling signals to a sparser grid.

Demonstration of features of the proposed MGA metrics and comparison with the state-of-the-art non-uniform distance metrics may also be completed with other signals, in particular when the instantaneous bandwidth allows for applying the unambiguous resampling model.

Determining the robustness of the metrics against various types of noise being an intrinsic part of each real-world signal falls beyond the scope of this paper but is another task to complete prior to application. In the experimental part, we already have two arbitrarily proposed factors: the shape of the sample environment (i.e., function $\phi$ in

Equation (14)) and the bandwidth-based local sampling interval series. Consequently, to maintain the research as repeatable and the description as general as possible, we postpone the noise-related analysis to future research.

*5.4. Advantages of the Mutual Graph Approximation*

The proposed Mutual Graph Approximation may be considered as an interesting trade-off between data distance accuracy and computational complexity. While the accuracy is sufficient to produce a reliable similarity criterion in most classification scenarios, the computational complexity remains acceptable for resource-limited implementations such as wearable systems.

Comparing the results of MGA (Tables 8 and 9) to the cross-correlation of signals resampled to the regular grid (Table 2), one may ask what the advantages are of comparing the signals directly in non-uniform sampling grids. One of the reasons is that the total complexity of such a process includes the complexity component from the cross-correlation (which is $O(L_1^2)$) and the other from the resampling process (i.e., $O(L_1^2 \cdot L_2)$ for cubic splines of $L_1$-points source data projected to $L_2$ target sampling points).

## 6. Conclusions

The technique of Mutual Graphs Approximation works as well as the most precise and less computationally demanding state-of-the-art distance measurement methods. Additionally, in applications such as signal classification and direct comparison of non-uniform signals, it may be equivalent to cross-correlation of resampled signals in terms of accuracy and computational cost. The obtained results show that the proposed MGA approach can be used as a simple but effective tool for assessing signal similarity straight in their non-uniform sampling grids.

**Author Contributions:** Conceptualization, P.A. and G.Ś.; methodology, P.A. and G.Ś.; software, P.A.; validation, P.A. and G.Ś.; formal analysis, G.Ś.; investigation, P.A. and G.Ś.; resources, P.A.; data curation, P.A.; writing—original draft preparation, P.A. and G.Ś.; writing—review and editing, P.A. and G.Ś.; visualization, P.A. and G.Ś.; supervision, P.A.; project administration, P.A.; funding acquisition, P.A. All authors have read and agreed to the published version of the manuscript.

**Funding:** This research was partly funded by AGH University of Krakow in 2023 as research project No. 16.16.120.773.

**Data Availability Statement:** The experimental part of the study is based uniquely on publicly available data sets referenced to as [40,41].

**Conflicts of Interest:** The authors declare no conflict of interest.

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
