# Peer review of "Quantifying Coincidence in Non-Uniform Time Series with Mutual Graph Approximation: Speech and ECG Examples"

_electronics, doi:10.3390/electronics12204228_

Round 1

Reviewer 1 Report

The approaches for direct comparison of time series in their original non-uniform grids are examined in the paper under review. The authors suggest using a linear graph to represent the non-uniform signal, and they employ the Mutual Graph Approximation (MGA) approach as a measure to determine how similar the patterns under consideration are to one another. Four cutting-edge techniques were used to create the MGA, and it was evaluated using ECG and speech signal examples projected to random and bandwidth-related grids. The reported findings demonstrate that the suggested MGA technique performs on par with the most precise and computationally less demanding state-of-the-art distance measures. Additionally, in applications such as signal classification, direct comparison of non-uniform signals might be equal to cross-correlation of resampled signals in terms of accuracy and computing cost. The obtained results show that the suggested MGA approach may be applied as a straightforward but efficient tool for evaluating signal similarity in non-uniform sample grids. Overall the content of the paper is interesting and deserves publication. The following are suggestions for improvements:

Line 26, revise "This means no á priori...".

Line 33, revise "but not thrifty nor justified" as "but neither thrifty nor justified".

I suggest citing the following unique book on the irregularity of graphs: A. Ali, G. Chartrand, P. Zhang, Irregularity in Graphs, Springer Cham, 2021.

In Eq. (4), use "\left(" and "\right)" to increase the height of the brackets, on the left side.

Correct Eq. (11).

In Figure, the label assigned to vertical axis is not clear.

In Table 9, please add some additional comments on the last two values of the second column (under avg.).

Optional suggestion: it would be nice to include the conclusion section. 

Reviewer 2 Report

This manuscript investigates the non-uniform time series with 

mutual graph approximation. The method of directly comparing time series in the original non-uniform grid is presented. The results indicate that the proposed MGA method can serve as a simple and effective tool. 

Reviewer 3 Report

The authors proposed a tool for scoring the similarity of signals in non-uniform sampling grids, called Mutual Graph Approximation (MGA)

Two types of dataset are used, speech signal and ECG. The proposed method shows better self-similarity score and good computing time compared to the state of the art.

On the other hand, although there is no good reference of dissimilarity, the distribution of similarity score difference may be used to compare the proposed method with other methods.
